# Using Sequence Analyses to Quantitatively Measure Oropharyngeal Swallowing Temporality in Point-of-Care Ultrasound Examinations: A Pilot Study

**DOI:** 10.3390/jcm13082288

**Published:** 2024-04-15

**Authors:** Wilson Yiu Shun Lam, Elaine Kwong, Huberta Wai Tung Chan, Yong-Ping Zheng

**Affiliations:** 1Department of Chinese and Bilingual Studies, The Hong Kong Polytechnic University, Hong Kong SAR, Chinahuberta.chan@connect.polyu.hk (H.W.T.C.); 2Research Institute for Smart Ageing, The Hong Kong Polytechnic University, Hong Kong SAR, China; yongping.zheng@polyu.edu.hk; 3Department of Biomedical Engineering, The Hong Kong Polytechnic University, Hong Kong SAR, China

**Keywords:** deglutition, dysphagia, aging, ultrasonography, information theory

## Abstract

(1) **Background**: Swallowing is a complex process that comprises well-timed control of oropharyngeal and laryngeal structures to achieve airway protection and swallowing efficiency. To understand its temporality, previous research adopted adherence measures and revealed obligatory pairs in healthy swallows and the effect of aging and bolus type on the variability of event timing and order. This study aimed to (i) propose a systemic conceptualization of swallowing physiology, (ii) apply sequence analyses, a set of information-theoretic and bioinformatic methods, to quantify and characterize swallowing temporality, and (iii) investigate the effect of aging and dysphagia on the quantified variables using sequence analyses measures. (2) **Method**: Forty-three participants (17 young adults, 15 older adults, and 11 dysphagic adults) underwent B-mode ultrasound swallowing examinations at the mid-sagittal plane of the submental region. The onset, maximum, and offset states of hyoid bone displacement, geniohyoid muscle contraction, and tongue base retraction were identified and sorted to form sequences which were analyzed using an inventory of sequence analytic techniques; namely, overlap coefficients, Shannon entropy, and longest common subsequence algorithms. (3) **Results**: The concurrency of movement sequence was found to be significantly impacted by aging and dysphagia. Swallowing sequence variability was also found to be reduced with age and the presence of dysphagia (*H*(2) = 52.253, *p* < 0.001, η^2^ = 0.260). Four obligatory sequences were identified, and high adherence was also indicated in two previously reported pairs. These results provided preliminary support for the validity of sequence analyses for quantifying swallowing sequence temporality. (4) **Conclusions**: A systemic conceptualization of human deglutition permits a multi-level quantitative analysis of swallowing physiology. Sequence analyses are a set of promising quantitative measurement techniques for point-of-care ultrasound (POCUS) swallowing examinations and outcome measures for swallowing rehabilitation and evaluation of associated physiological conditions, such as sarcopenia. Findings in the current study revealed physiological differences among healthy young, healthy older, and dysphagic adults. They also helped lay the groundwork for future AI-assisted dysphagia assessment and outcome measures using POCUSs. Arguably, the proposed conceptualization and analyses are also modality-independent measures that can potentially be generalized for other instrumental swallowing assessment modalities.

## 1. Introduction

Normal deglutition, or swallowing, is a physiological process that requires well-timed coordination and delicate control of over 30 pairs of oropharyngeal muscles to achieve safe (airway protection) and efficient nutritional intakes. It is usually conceptualized by a stage model that involves four phases: (i) The oral preparatory phase, where bolus formation (for solid) and manipulation (for both liquid and solid) take place while the lip seal and posterior tongue act as gates to prevent anterior and pre-mature spillage. This is followed by (ii) the oral (transit) phase, where the bolus is propelled posteriorly to the oropharynx, and (iii) where the oropharyngeal swallow (the pharyngeal phase) is triggered to facilitate airway protection, enforced by hyolaryngeal excursion with epiglottic inversion and laryngeal vestibule closure, and swallowing efficiency, which is enforced by pharyngeal squeeze with the upper esophageal sphincter (UES) opening and velopharyngeal closure to prevent nasal regurgitation. Finally, there is (iv) the esophageal phase, where the bolus is moved by esophageal peristalsis to the digestive system [1]. Compromise to the integrity of this mechanism may result in swallowing disorders, or dysphagia, which can lead to malnutrition, dehydration, aspiration pneumonia, and suffocation [1].

### 1.1. Swallowing Assessment Modalities

The existing gold standards of dysphagia diagnosis involves the use of instrumental assessments, namely a videofluoroscopic swallowing study (VFSS) and fiberoptic endoscopic evaluation of swallowing (FEES). Validated and standardized assessment and rating protocols like Modified Barium Swallow Impairment Profile (MBSImP) [2] (for VFSS) and Penetration-Aspiration Scale (PAS) [3] (for both VFSS and FEES) are also available for clinical usage. Recently, these tools have also been incorporated into the multidisciplinary management protocol of swallowing disorders of different etiologies, e.g., FEES for amyotrophic lateral sclerosis [4]. Nevertheless, despite their merits and well-established evidence, these modalities are either radioactively or physically invasive, could only be accessed at specific medical settings, and require certain levels of patient cooperation and compliance to implement.

In response to the issue of invasiveness, recent studies have also begun to explore the use of (surface) electromyography (sEMG) as a non-invasive alternative, and various protocols have been proposed to examine and understand the oral, pharyngeal, and laryngeal muscle activities during swallowing, particularly for neurological dysphagia [5]. For example, sEMG has been used to examine the symmetry of muscle activities among people with Parkinson’s disease [6]. However, as discussed in [5], electromyographic examinations of swallowing muscle activities are prone to impedance noise from the skin surface, and the relationship between muscle force and sEMG signal amplitude requires further investigations. In other words, sEMG might not be able to completely offer task-specific information about one’s swallowing physiology.

On the other hand, ultrasonography, particularly a point-of-care ultrasound (POCUS), is a non-invasive, task-specific, and relatively accessible modality that can even be conducted on infants [7], children with physical disabilities [8], and possibly people with impaired cognitive abilities, requiring little manpower and time constraints. Clinically, this is particularly important when working with a variety of populations in primary care, outreaching, and telepractice settings. To date, in addition to the absence of standardized and validated protocols for acquisition/operation using POCUS for swallowing assessments, there is a lack of standardized, clinically significant, and relevant parameters to be extracted and observed from ultrasonographic swallowing examinations. Hence, it is essential to explore a set of reliable and valid quantitative measurement techniques for extracting clinically relevant parameters in the POCUS swallowing examinations.

### 1.2. Swallowing Sequencing: The Status Quo

While previous studies utilizing ultrasonography have largely focused on quantitative measurements of swallowing kinematics of key anatomical structures, e.g., hyoid bone displacement (e.g., [9,10,11,12]), tongue base retraction (e.g., [9,13,14,15]), and geniohyoid muscle contraction (e.g., [12]), recent attempts have been made to use ultrasonography to study temporal profiles of swallowing event sequences (e.g., [16]). As Mendell and Logemann [17] have argued, the temporal organization of a swallowing event offers essential information about the effectiveness of swallowing, and therefore its measurement adds an additional and important dimension to gauge and characterize the nature of swallowing physiology.

Most studies investigating swallowing temporal sequences have so far been using adherence, i.e., matching the actual sequences with anticipated sequences by observing the frequency or computing matching percentage, as the obligatory sequence identification and variability measurement (e.g., [16,17,18,19,20]). Herzberg and colleagues [20] found the obligatory subsequence of laryngeal elevation followed by UES opening, followed by hyolaryngeal approximation from healthy young and old swallows using VFSS. Their adherence results indicated reduced variability in swallowing sequences among aging individuals by comparing adherence across trials [20]. Kwong and colleagues [16] established the obligatory subsequence of hyoid bone displacement onset, followed by tongue base maximum retraction, followed by hyoid bone displacement offset from six swallowing events extracted from healthy swallows acquired through ultrasonography. They also found that bolus type might have an effect on swallowing event adherence [16]. As we shall discuss in Section 1.4.2 and Section 1.4.3, although adherence reveals useful local information about event sequences (i.e., focusing on a pair of events at a time), the study of obligatory sequences and variability could conceivably adopt a more global, systemic, and algorithmic approach.

### 1.3. Ultrasound Swallowing Assessment and Quantitative Medical Image Analysis

Meanwhile, it has been well-noted that quantitative and medical image analysis, particularly frame-by-frame analysis of ultrasound images and identification of key swallowing events, could be labor-intensive and prone to inconsistency and human errors, thus hindering the translation to clinical practice in the long run. While leveraging machine learning (ML) in assisting the examination and analysis process could be a viable solution, maintaining transparency, or interpretability, of machine learning models is essential to establish clinical acceptance and offer insights into management [21]. It is also important to establish evidence-based, clinician-crafted, or understandable features while setting off for the exploration of utilizing POCUSs as a swallowing assessment modality and ML for automated analysis.

### 1.4. The Proposed Framework: A Systemic Conceptualization

The current study proposed and adopted a systemic conceptualization of the oropharyngeal phase of the deglutition process. This overall conceptualization takes a bottom-up approach by delineating swallowing sequencing from individual instance, individual systems, to population levels.

#### 1.4.1. Swallowing Sequence Instances

The swallowing process, as we have seen above, has long been considered as stages of multiple movement events. In light of the four stage model [1], Logemann [22] considered the deglutition mechanism as a series of well-timed and adequate opening and closing of a set of tubes and valves. For each swallow, given that the opening and closing events of the ‘tubes’ and ‘valves’ commence at discrete time points, these events also form a swallowing event sequence. Conceivably, to achieve swallowing safety and efficiency, the coordination of each swallowing anatomical structure shall require (partial) overlaps in their opening and closure duration in a single swallow. In other words, to extend Logemann’s [22] idea, a single swallowing instance can be seen as a gross process which comprises concurrent subprocesses (i.e., a parallel movement timeline of each individual structure, like a Gantt chart) (Conceptualization #1 in Figure 1). Meanwhile, as the movement of each individual anatomical structure takes place with respect to time, an ordered sequence of distinctive swallowing movements can also be observed. The sequence of these distinct movements can conceivably be simplified and encoded as pseudo-DNA sequences like many other biological sequences (Conceptualization #2 in Figure 1).

#### 1.4.2. Swallowing Sequencing System

With respect to swallowing sequences and their variability, previous studies often adopted a ‘local’ view of the swallowing event sequence by checking the adherence of the executed event pairs against certain expected sequences, yielding ‘obligatory sequence(s)’ by 100% adherence and variability by statistical tests [16,17,18,19,20]. We extended such a perspective to a more ‘global’ level by considering each swallowing sequence as an instantiation from an individual’s swallowing system, which could potentially be a distribution of different sequences (i.e., movement events in different orders) (Conceptualization #3 in Figure 1).

#### 1.4.3. Obligatory Swallowing Sequences

By examining all executed swallowing sequences of a particular group, we could also gain insights of common sequences that characterize the group’s swallowing mechanism. Given that the swallowing sequence has been conceptualized and encoded as a form of biological sequence, identifying an ‘obligatory sequence’ in turn requires finding the longest common subsequence (LCS) from a set of swallowing event order sequences.

LCSs are a longstanding and well-defined problem in computer science and bioinformatics. As the name suggests, an LCS is a subsequence that is common to a set of ordered sequences. Consider the following simplified example of a set of three sequences:

Sequence 1—A-**B**-C-**D**-**E**-**F**

Sequence 2—F-**B**-A-**D**-**E**-C

Sequence 3—F-A-**B**-**D**-C-**E**

The subsequence B-D-E (highlighted above) is considered as the LCS, as it appears in all of the above sequences, despite other letters appearing in between. While A-F and A-C are also common subsequences, they are not the longest when compared to B-D-E. In this example, the LCS is obtained from more than two sequences; this LCS is also known as a multiple longest common subsequence (MLCS). In the case of Kwong and colleagues [16], the subsequence of hyoid bone displacement onset, followed by tongue base maximum retraction, followed by hyoid bone displacement offset, would be perceived as the MLCS obtained from ultrasonography. Accordingly, the obligatory sequence of laryngeal elevation, followed by UES opening, followed by hyolaryngeal approximation in [20], is also an MLCS of oropharyngeal swallows obtained from VFSS examinations.

### 1.5. Research Objectives

This study aimed to (i) propose a systemic conceptualization of swallowing physiology; (ii) quantify and characterize the temporality of swallowing event sequences, including sequence concurrency and variability of different groups and obligatory sequence among healthy young adults, using sequence analyses; and (iii) gauge the effect of aging and dysphagia on the quantified temporal features. Findings of the present study will provide preliminary evidence of the applicability of sequence analyses to ultrasonographic swallowing examinations and lay the groundwork for further development and validation of ML-integrated swallowing examinations using POCUSs.

## 2. Materials and Methods

### 2.1. Participants

The study was conducted in accordance with the Declaration of Helsinki and approved by the Institutional Review Board of the Hong Kong Polytechnic University (Ref. No: HSEARS20210607002). Forty-three participants, including (i) 17 healthy adults (mean age = 41.18 years, SD = 17.37), (ii) 15 healthy older adults (mean age = 69.46 years, SD = 5.45), and (iii) 11 dysphagic adults (mean age = 63.55 years, SD = 12.32), were recruited from the community, daycare centres, and nursing homes for the elderly in Hong Kong from July 2021 to October 2021. Detailed demographics are summarized in Table 1 below.

The non-dysphagic subjects aged below 65 and those 65 or above were assigned to the Healthy Young Adult and Healthy Older Adult Group, respectively. They reported no previous history of swallowing difficulties, nor any etiologies known to be associated with dysphagia. The 3-ounce water screening test [23] was administered to rule out possible under-diagnosed dysphagia in these two groups. Subjects in the dysphagic group were previously diagnosed with oropharyngeal dysphagia by a qualified speech language therapist through clinical bedside examinations and/or instrumental assessment(s). They were recommended in regard to food texture and fluid consistency with reference to the International Dysphagia Diet Standardization Initiatives (IDDSI) [24]. None of them had undergone surgical removal of swallowing-related head and neck structures. All participants were able to follow simple commands in Cantonese.

### 2.2. Materials and Equipment

Ultrasound swallowing images (USIs) were acquired with a curvilinear ultrasound transducer (Mode AC2541 by Esaote) connected to a portable ultrasound system (MyLabGamma by Esaote, Genova, Italy). An Action^®^ BOL-X-I gel pad with film of dimensions 10 cm × 10 cm × 1 cm was fitted to the submental and neck regions to avoid air gaps and thus obtain optimal images (see Figure 2 for illustration of experimental setup). All thickened liquid was prepared with respect to the IDDSI framework [24] using gum-based thickener (Senior Deli Clear Thickener, CareEZ Senior Deli, Hong Kong SAR, China).

### 2.3. USI Acquisition

Subjects sat comfortably upright or in the recommended feeding position, with head and trunk support from a wall, a positioning aid, or manual support. All USIs were acquired by a final-year Master of Speech Therapy student (the third author H.C.) who had undergone training on dysphagia management and at least 10 supervised hours of ultrasound imaging operations. The acquisition of USIs was conducted using B-mode ultrasound on the mid-sagittal plane of the submental region using a frame rate fixed at 15 Hz, frequency at 8 MHz, depth of 108 mm, dynamic range at 11, and density and persistence at 3. Figure 2b shows the placement of the transducer without a gel pad. The footprint of the images spanned anteriorly−posteriorly from the mental protuberance of the mandible (reflective) to the thyroid cartilage (reflective), while the tongue base, geniohyoid muscle, and hyoid bone were visible in the scope of view. An annotated sample frame of USI is shown in Figure 2c.

### 2.4. Swallow Trial Protocol

Subjects were instructed to perform the following swallowing tasks: (i) dry swallows, and (ii) swallowing of 5 mL and 10 mL of IDDSI Level 0 to 4 liquid boluses, performing each task (a single swallow) for five repeated and discrete trials. For each trial, a short hum (/m/) was performed before and after swallowing the bolus. Vocal fold vibrations resulted from humming, which could be observed at the thyroid shadow, helped mark the beginning and the end of the swallowing task. This provided a quick reference to the Frames of Interest (FOIs) in timestamp annotations and a landmark for relative referencing to the Structures of Interest (SOIs). For task (ii), subjects were fed using a syringe. In case of fatigue and/or other issues (e.g., restrictions of intake volumes), subjects only performed the tasks for three trials, while the volume of the water bolus could be altered to 3 mL. Dysphagic subjects only consumed the fluid consistency and volume recommended by their respective speech language therapists in task (ii). A schematic representation of the protocol is illustrated in Figure 3.

### 2.5. Data Annotation

From the recorded USIs, frames of the respective onset, maximum, and offset of the SOI movement events, namely hyoid bone displacement (HB), tongue base retraction (TB), and geniohyoid muscle contraction (GH), were identified by two raters, a final-year Master of Speech Therapy student (Rater 1, also the third author H.C.) and research personnel with speech and language science backgrounds and annotation training for at least 10 h (Rater 2). Following the definitions in [16], the event states were identified as follows:Event onset: the first frame where the SOI begins leaving its at-rest position, thus the commencement of the movement (for onset of HB, commencement of anterior-superior displacement was considered);Event maximum: the first frame where the SOI reaches its maximum displacement or contraction;Event offset: the frame at which the SOI begins returning to the at-rest position.

Considering that the frames are on a continuous scale with fixed intervals, the agreement and consistency of the annotations were computed using a two-way mixed effect and random effect intraclass correlation coefficient (ICC), respectively. The raters achieved excellent inter-rater reliability (ICC = 0.98) and intra-rater reliability (Rater 1 ICC = 0.99, Rater 2 ICC = 0.91).

One thousand four hundred and fifty-one out of 1568 annotated USIs were selected for further analysis, and 117 were discarded due to substantial artifacts and/or missing key SOIs (e.g., out of sight), including 825 from healthy young adults, 547 from healthy elderly, and 79 from dysphagic individuals.

For presentation simplicity, the following notations will be adopted hereafter:An event state is denoted with subscripts: e.g., HB_on_ as the onset, HB_max_ as the maximum, and HB_off_ as the offset of hyoid bone displacement.The duration from onset to maximum was defined as the initiating period using subscripted init and that from maximum to offset as the sustaining period using subscripted sus: e.g., HB_init_ denotes the onset to maximum duration/transition and HB_sus_ the maximum to offset duration/transition of hyoid bone displacement, respectively.The overlapping transitions of two events are denoted by hyphenating the abbreviations of the events with a subscripted abbreviation of transition: for instance, HB-TB_init_ denotes the overlapping period of the onset-to-maximum transitions of hyoid bone displacement and tongue base retraction.

#### 2.5.1. Data Preprocessing

The annotated timestamps were first sorted in ascending order. Identical timestamps within the same sequence were further permuted in raw annotations to capture all possible event sequences (See Figure 4a for sample visualization). Formally, for an ordered swallowing sequence S = {s_1_, s_2_ … s_n_|s_1_ ≤ s_2_, … s_n−1_ ≤ s_n_}, permuted ordered sequences were generated by permuting the subsequence {s_i_, s_j_, s_k_} ⊆ S if s_i_ = s_j_ = s_k_. After permutations, 8309 possible sequences were identified in healthy adults, 3281 in healthy elderly people, and 249 in dysphagic individuals. A visual presentation of a sample permutation has been shown in Figure 4a.

### 2.6. Data Extraction and Statistical Analysis

Unless otherwise specified, all data pre-processing and extraction, algorithms, and statistical analysis were performed/implemented in custom codes developed in Python 3.9.9 and executed in Jupyter Lab.

#### 2.6.1. Concurrency Analysis: Overlap Coefficient

Recall that a swallow can be seen as a gross process with subprocesses whose event states (i.e., onset, maximum, and offset) and periods (i.e., initiating and sustaining) occur (a)synchronously in the same temporal dimension at the instance level (Conceptualization #1 in Figure 1 and swallowing instances in Figure 4b). In order to capture the degree to which the duration of the events overlap with one another, a pairwise Overlap Coefficient, as known as the Szymkiewicz−Simpson overlap coefficient (SSC) (Equation (1)), was computed across the event period durations, e.g., the overlapping onset-to-maximum duration of hyoid bone displacement and tongue base retraction (i.e., HB-TB_init_), for each swallowing event sequence:(1)$$SSC\left(P,Q\right)=\frac{\left|P\cap Q\right|}{\min\left(\left|P\right|,\left|Q\right|\right)}$$
where *P* and *Q* are two sets of distinctive event period timestamp. If *P*
⊆
*Q* or *Q*
⊆
*P*, SSC = 1.0.

Figure 5 illustrates an annotated sample extract of 15 frames (equivalent to 1 s) from a healthy young participant. In the sample of Figure 5, the onset-to-maximum period of hyoid bone displacement (HB_init_) spans from frames 173 (HB_on_) to 179, and the maximum-to-offset period ranges (HB_sus_) from frames 180 (HB_max_) to 182 (HB_off_). Hence, HB_init_ forms the set {173, 174, …, 178, 179} and HB_sus_ forms the set {180, 181, 182}. Accordingly, the onset-to-maximum period of geniohyoid muscle contraction (GH_init_) spans from frames 176 (GH_on_) to 179, and maximum-to-offset (GH_sus_) from frames 180 (GH_max_) to 182 (GH_off_); thus GH_init_ equals {176, 177, 178, 179} and GH_sus_ equals {180, 181, 182}. The SSC of HB_init_ and GH_init_ (i.e., HB-GH_init_) is then computed as follows:$$\mathrm{SSC}({\mathrm{HB}}_{\mathrm{init}},\;{\mathrm{GH}}_{\mathrm{init}})=\frac{\left|{HB}_{init}\cap {GH}_{init}\right|}{\min\left(\left|{HB}_{init}\right|,\left|{GH}_{init}\right|\right)}=\frac{\left|\{176,177,178,179\}\right|}{\min\left(6,4\right)}=\frac{4}{4}=1.0$$

Following the same computation procedure, it can also be shown that the SSCs of all other overlapping periods (i.e., TB-HB_init_, TB-GH_init_, HB-GH_sus_, TB-HB_sus_, TB-GH_sus_) are also 1.0. Visually, as shown in the Gantt chart in the middle of Figure 5, it is also evident that when the timestamp of a shorter event period (e.g., GH_init_) is a subset of a longer event period (e.g., HB_init_), meaning that the shorter event period takes place concurrently within the timeframe of its longer counterpart, the SSC will legitimately be 1.0.

SSC has been used as an evaluation metric of similarity between gene expression networks in some bioinformatic studies (e.g., [25]). While other similarity measures, such as Jaccard Index and Dice Coefficients, are more common in medical and bioinformatic research, SSC was chosen as it indicates a total overlap (i.e., an SSC of 1.0) when an event transition occurs fully within the time frame of another event transition. For the current study, SSCs were computed for the set of timestamps of all 1451 swallows after removing invalid trials. Specifically, SSCs of the swallows across different groups were compared to investigate the effect of aging and dysphagia of event concurrency.

#### 2.6.2. Variability Analysis: Shannon Entropy

Given the conceptualization that a swallow could be deemed as an instantiation of one’s swallowing mechanism, which could further be conceived as a probability distribution of different swallowing event sequences (Conceptualization #3 in Figure 1 and individual system in Figure 4b), like drawing a marble from an urn, the variability of swallowing event sequences from the same bolus type could be quantified using Shannon entropy (Equation (2)), which has found extensive application in computational biology [26]:(2)$$H\left(m_{i}\right)=-\sum_{j}p\left(m_{i}^{j}\right)logp(m_{i}^{j})$$
where $p\left(m_{i}^{j}\right)$ is the probability of the *j*-th swallow sequence of the *i*-th unique bolus type $m_{i}$ with repeated trials, given a group of aggregated swallowing trials $M=\left\{m_{1}\ldots m_{k}\right\}$, and where *k* is the total number of aggregated swallowing trials of the same bolus type by different individuals. Intuitively, Shannon entropy quantifies the amount of uncertainty, hence variability, of a system (here it refers to an individual’s swallowing sequencing system).

As illustrated in the diagram at the bottom of Figure 5, a single swallow trial may incur more than one possible sequence. Recall that this is due to the application of permutation (described in Section 2.5.1), which treats identical timestamp equally likely in a probability distribution. Hence, when aggregating with other trials of the same bolus type, the total number of possible sequences could exceed the number of trials.

For the current study, the sequence of swallow trials of the same bolus volume and consistency were grouped together to form 345 groups of aggregated swallows, and their respective variability were analyzed using Shannon entropy. Comparisons were first made with respect to bolus types that were common to all groups (*n* = 199) to compare the variability of the swallowing sequence on the same set of bolus types. This was followed by comparisons of all 345 groups of aggregated swallows to compare the variability for all possible sequences given different bolus types with a larger sample size.

#### 2.6.3. Obligatory Sequence: Multiple Longest Common Subsequence

As previously discussed, an obligatory swallowing sequence could be redefined as the LCS that all swallowing sequences adhere to (i.e., 100% adherence), often in healthy swallows to characterize the nature of normal deglutition [18]. Given that swallowing sequences are encoded as a sequence of DNA-like alphabetical letters (Conceptualization #2 in Figure 1), finding all obligatory sequences from multiple swallowing event sequences is in turn finding the MLCS, which has been used by bioinformaticians for multiple sequence alignment in the study of DNA sequences [27].

For the current study, a leveled directed acyclic graph (DAG) MLCS algorithm [28] was implemented in Python and applied to all 8309 possible swallowing event sequences to obtain obligatory sequence(s) of healthy swallows. Essentially, the leveled DAG algorithm takes recurrent steps to identify the MLCS. At each step, it considers multiple partial LCSs (i.e., common subsequences), removes non-LCS subsequences, and runs until all possible partial LCSs are explored. This algorithm was adopted due to its computational efficiency; for more technical details about the algorithm, readers are recommended to refer to [28]. In addition, to compare the current results of MLCS with previous studies, a cross tabulation of event order probability (i.e., adherence) among event states was also constructed to indicate adherence.

#### 2.6.4. Statistical Analysis

Except for MLCSs, descriptive statistics were performed for all quantified variables resulting from sequence analysis. With respect to Objective (iii) of the present study, Group was considered as the independent variable. For inferential statistics, Kruskal–Wallis *H* Tests were conducted to examine the effect of the independent variables on the respective sequence feature, followed by post hoc Dunn’s tests to examine if statistical significance (i.e., *p* < 0.05) was indicated. Open-source Python libraries were used to perform the abovementioned analyses: pandas [29] for data preparation and descriptive statistics, matplotlib [30] and seaborn [31] for data visualization, and pingouin [32] and scikit-posthocs [33] for inferential statistics.

## 3. Results

### 3.1. Concurrency Analysis: Overlap Coefficient (SSC)

Figure 6 shows the means of the overlap coefficient of paired events across different groups. Highly concurrent pairwise movements across different state transitions were exhibited in the healthy young adults, compared to two other groups. As summarized in Table 2, there was a significant difference for all concurrent movement pairs. A large effect size was indicated in HB-GH_init_ (*H*(2) = 622.7, *p* < 0.0001, η^2^ = 0.429), TB-HB_init_ (*H*(2) = 526.413, *p* < 0.0001, η^2^ = 0.362), HB-GH_sus_ (*H*(2) = 456.167, *p* < 0.0001, η^2^ = 0.314) and TB-HB_sus_ (*H*(2) = 571.85, *p* < 0.0001, η^2^ = 0.394), a moderate effect size in TB-GH_init_ (*H*(2) = 99.907, *p* < 0.0001, η^2^ = 0.068), and a small effect size in TB-GH_sus_ (*H*(2) = 16.745, *p* < 0.001, η^2^ = 0.011). Post hoc analyses confirmed differences in TB-GH_init_ and TB-HB_init_ across healthy young adults, healthy older adults, and dysphagic individuals, differences in HB-GH_init_, HB-GH_sus_, and TB-HB_sus_ between healthy young adults and the other two groups, as well as TB-GH_sus_ between dysphagic and the other two groups (see Table 2).

### 3.2. Variability Analysis: Shannon Entropy

As shown in Table 3, swallowing sequences of the healthy young adults exhibited the highest variability (mean entropy = 5.119 ± 1.095), followed by the healthy older adults (mean entropy = 4.428 ± 0.95) and the dysphagic adults (mean entropy = 3.038 ± 0.872) upon matching the common bolus type (i.e., the same bolus volume and consistency). A Kruskal–Wallis *H* test revealed a statistically significant effect of groups with large effect size (*H*(2) = 52.253, *p* < 0.0001, η^2^ = 0.260). Post hoc Dunn’s tests confirmed differences among all the groups (*p* < 0.001 for healthy young adults vs. healthy older adults, and *p* < 0.0001 for all other pairwise comparisons).

The variability of the aggregated swallowing event sequence across different bolus types in different groups was further compared. Similar to the results of the matched bolus type variability, high variability was indicated in the swallowing sequences in the healthy young adults (mean entropy = 5.07 ± 1.075), followed by the healthy older adults (mean entropy = 4.286 ± 1.095), and subsequently by the dysphagic adults (mean entropy = 2.896 ± 1.011). The effect of the group was statistically significant with a large effect size (*H*(2) = 78.472, *p* < 0.0001, η^2^ = 0.224), and the differences among the three groups were confirmed by post hoc Dunn’s tests (*p* < 0.0001).

### 3.3. Obligatory Sequence: Multiple Longest Common Subsequence

Among the 8309 possible (permuted) swallowing event sequences obtained from the healthy young adults (*n* = 17) swallowing water boluses of various volumes and consistencies (total number of swallows = 825), 336 unique sequences were identified. Four non-trivial obligatory event sequences were identified from the MLCS algorithm:TB_on_ prior to GH_max_ prior to GH_off_ (Sequence 1);HB_on_ prior to GH_max_ prior to GH_off_ (Sequence 2);GH_on_ prior to HB_max_ prior to HB_off_ (Sequence 3);TB_on_ prior to HB_max_ prior to HB_off_ (Sequence 4).

Here, a trivial event sequence refers to the transition from merely the onset to maximum and from maximum to offset of the same structure, e.g., HB_on_ prior to HB_max_ prior to HB_off_, and non-trivial event sequences should include event ordering of different structures of different states, e.g HB_on_ prior to TB_max_ prior to TB_off_.

To visualize and exemplify the obligatory sequences in ultrasound images, Figure 7 presents images annotated using event labels (top), the possible sequences (middle), and the identified MLCSs (bottom) in the same sample video extract as Figure 5. As shown in the sequence profiles in the middle of Figure 7, all possible sequences after permutation adhered to the above obligatory sequences. Readers can also refer to Appendix A in the Appendix A for annotated images of each movement in motion.

Alternatively, obligatory sequences could be revealed by examining pairwise event order probability. Table 4 further summarizes the event order probability (i.e., adherence) of all ordered event pairs among 8309 possible sequences found in healthy young adults. The first and the second event are presented on the vertical and the horizontal axis, respectively. An event order probability of 1.0 indicates that the first event was always followed by the second event, and, accordingly, 0.0 indicates the first was never followed by the second. The event order probabilities, or adherence, of an event pair from both directions (e.g., HB_on_ prior to GH_on_, and GH_on_ prior to HB_on_) shall always add up to 1.0. The above non-trivial sequences could also be obtained and validated by checking against the pairs with an event order probability of 1.0 in Table 4.

In addition to the obligatory sequences, high adherence of two previously reported pairs was also indicated: (i) 99.98% adherence (8307/8309) for HB_on_ prior to TB_max_, and (ii) 98.58% adherence (8191/8309) for TB_max_ prior to HB_off_ [16].

## 4. Discussion

The present study aimed to (i) propose a systemic conceptualization of oropharyngeal swallows in the deglutition process; (ii) quantify and characterize swallowing movement temporality using overlap coefficients (concurrency), Shannon entropy (variability), and multiple longest common subsequences (obligatory sequences); and (iii) investigate the effect of aging and dysphagia on these quantitative measures. Overall, the results showed that the concurrency and variability of oropharyngeal swallows could be impacted by normal ageing and the presence of dysphagia. Four obligatory sequences were also found to be exclusive to the healthy young individuals. These findings might further our knowledge about, and hence the measurement of, the age-related changes and impairments of the swallowing physiology, which we shall discuss in the sections below.

### 4.1. Concurrency

One key observation from concurrency analysis was that there were three types of concurrent movement pairs that could possibly differentiate (i) healthy young adults from healthy older adults and dysphagic adults, (ii) dysphagic from non-dysphagic adults (i.e healthy young and older adults), and (iii) the three groups of individuals from one another.

Firstly, three sets of concurrent movements differentiated healthy young adults from healthy older and dysphagic adults, namely (i) the onset-to-maximum overlaps of hyoid bone displacement and geniohyoid muscle contraction (HB-GH_init_), (ii) the maximum-to-offset overlaps of hyoid bone displacement and geniohyoid muscle contraction (HB-GH_sus_), and (iii) the maximum-to-offset overlaps of TB and HB (TB-HB_sus_). Significantly fewer overlaps in the onset-to-maximum and maximum-to-offset of HB and GH among the healthy older and dysphagic adult groups may suggest that the whole hyoid bone displacement and geniohyoid muscle contraction sequences might have been partially dissociated from each other. This may also imply that different muscles (e.g., mylohyoid which displaces the hyoid bone superiorly [34]) might be engaged to move the hyoid bone to compensate for reduced strength and/or control of the geniohyoid muscle. This physiological difference may also be due to possible age-related mass and quality changes in the geniohyoid muscle [35] and other associated conditions that impact the control of suprahyoid muscles. While falling out of the scope of the present study, this hypothesis may be further verified by observing and correlating HB-GH_init_ and HB-GH_sus_, as well as the mass and quality of the geniohyoid and other suprahyoid muscles with the magnitude of anterior and superior hyoid bone displacements.

A similar dissociation pattern was also observed for the maximum-to-offset overlaps of tongue base retraction and hyoid bone displacement (i.e., TB-HB_sus_). This in turn suggested that part of the bolus transit process (driven mainly by tongue base retraction), as well as airway protection and cricopharyngeal (CP) opening (driven mainly by hyoid bone excursion), briefly worked with each other when they reached and sustained the maximum retraction/displacement. As we shall discuss in later sections, such physiological changes or differences from healthy young adult deglutition could also be seen in the adherence of event pairs.

Secondly, the maximum-to-offset overlaps of tongue base retraction and geniohyoid muscle contraction (i.e., TB-GH_sus_) differentiated healthy young and older adults from their dysphagic counterparts. In terms of muscle control, this may imply that both healthy adults, regardless of age, tended to engage tongue base retraction for bolus transition and geniohyoid muscle contraction to promote hyoid bone displacement and CP opening. Nonetheless, this was not likely to be the case for dysphagic individuals. Interestingly, this also lent further support to the above hypothesis that the role of geniohyoid muscles in sustaining hyoid bone displacement among healthy older adults would have been compensated by other suprahyoid muscles, as we could observe that TB-HB_sus_ tended to dissociate while TB-GH_sus_ did not. On the other hand, dysphagic individuals might have further difficulties to utilize the geniohyoid muscles (and possibly other suprahyoid muscles) to work with tongue base retraction for sustaining bolus transit and airway protection due to their respective neurological/structural impairments.

Finally, two initiating concurrent movements, namely the onset-to-maximum overlaps of (i) tongue base retraction and geniohyoid muscle contraction (i.e., TB-GH_init_) and (ii) tongue base retraction and hyoid bone displacement (i.e., TB-HB_init_), differentiated the three groups from one another. A general pattern where the concurrency of these movements decreased with normal aging and further deteriorated with neurological/structural impairments to the deglutition mechanism was also observed. This also echoed with the hypotheses above that aging swallows were characterized by a partial dissociation between the movements of HB and GH with possible compensation, while dysphagic swallows were characterized by partial dissociations among the movements HB, GH, and TB, with possible physiological constraints to compensate for the respective impairments.

With respect to the analytical method adopted, despite a lack of reference to previous research and not being conclusive, concurrency analysis using an overlap coefficient seems to offer an additional dimension to the study of swallowing event sequences and kinematics. It also characterizes minute differences among different groups. It should be, however, carefully noted here that ‘concurrency’ only refers to the overlapping duration between *two* paired swallowing movements, e.g., the duration of the tongue base and geniohyoid muscles; it is not an overall measurement of overlaps of all the swallowing events of concern.

### 4.2. Variability

The present study found significantly higher variability in events related to tongue base retraction, hyoid bone displacement, and geniohyoid muscle contraction in healthy young swallows than in aging and dysphagic swallows. This echoes the results indicated in previous studies using videofluoroscopy. Herzberg and colleagues [20] found significantly reduced variations in 5 mL thin and 20 mL thickened liquid among healthy older adults using the proportion of unique sequences identified from the aggregated number of sequences, attributing the variation to pharyngeal transit time in videofluoroscopic images. While the present study did not include pharyngeal contraction and transit as a movement parameter, the imaging and inclusion of geniohyoid muscles offers an additional perspective to account for possible age-related changes to the variability of swallowing event sequencing. Similar to the explanation for reduced concurrency of movement events, reduced variability in swallowing event sequences among healthy older adults might also be explained by reduced mass, which leads to reduced contraction force in the swallowing muscles [35], thus restricting the flexibility of movement sequencing. Nonetheless, since the present study did not include the analysis of muscle mass and quality, further validation is warranted to confirm the relationship between age-related muscle changes (e.g., sarcopenia) and sequence variability.

Interestingly, when the healthy older adults and dysphagic adults were compared, the former still exhibited higher variability than the latter. This resonates with the possibility that the modulation, execution, and coordination of sensorimotor responses during voluntary swallows were further impaired in dysphagic individuals, e.g., as indicated previously among post-stroke patients [36]. On the other hand, healthy older adults might have compensated for restrictions from reduced muscle mass through other muscles so that their swallowing movement sequence remained relatively more varied than the dysphagic individuals’. This in turn suggests that anatomical/physiological changes or constraints imposed by aging or the etiology of dysphagia might not only be reflected in reduced concurrency, but also in reduced variability of swallowing movement event sequences. Further investigation, nonetheless, is again warranted to confirm such a hypothesis of compensatory mechanisms among healthy older adults and the possible effects of age-related (e.g., sarcopenia) muscle changes.

Grossly speaking, the current results of variability measures echo the findings from previous studies and other measurements of event sequences, such as concurrency, and hence provides preliminary support for applying Shannon entropy as a form of sequence analysis for swallowing event sequence variability, as well as the conceptual proposal that each swallow sequence is an instantiation of one’s swallowing sequencing system. It should be noted that the current study only included a limited sample size and that the validity of Shannon entropy, although promising, needs to be further verified.

### 4.3. Obligatory Pairs in Healthy Young Adults

The current study also examined obligatory sequences among healthy young adults at a macro level to reveal common physiological event sequences of the deglutition mechanism independent of any impairments and the possible effects of ageing. With permutations of identical timestamps in the swallowing event sequences, four non-trivial obligatory sequences were identified, namely:(i)Tongue base retraction onset prior to geniohyoid muscle contraction maximum, prior to geniohyoid muscle contraction offset (Sequence 1);(ii)Hyoid bone displacement onset prior to geniohyoid muscle contraction maximum, prior to geniohyoid muscle contraction offset (Sequence 2);(iii)Geniohyoid muscle contraction onsets prior to hyoid bone displacement maximum, prior to hyoid bone displacement offset (Sequence 3);(iv)Tongue base retraction onsets prior to hyoid bone displacement maximum, prior to hyoid bone displacement offset (Sequence 4).

These findings are in line with the general understanding of the normal deglutition process. Sequences 1 and 4 suggest that, to achieve swallowing efficiency, bolus propulsion must always commence before complete airway closure, which is achieved partially by maximum hyoid bone displacement and maximum geniohyoid muscle contraction [37,38]. Interestingly, as also shown in Table 4, the completion of initiating and sustaining both hyoid bone displacement and geniohyoid muscle contraction were not always further interjected with a tongue base retraction maximum, let alone its offsets. Although Table 4 shows that tongue base retraction maximum was reached before the offsets of hyoid bone displacement (98.58% adherence) and geniohyoid muscle contraction (97.53% adherence) most of the time, the findings of Sequences 1 and 4 imply that the airway protection mechanism, partially driven by hyoid bone displacement, did not always sustain at its maximum when bolus propulsion was still initiating/sustaining among healthy young adults.

Previous studies relied on post-mortem examinations [37] and approximating the positions of suprahyoid muscles in VFSS images [34] to determine the role of the geniohyoid muscle as a member of the anterior muscular sling that pulls the hyoid bone forward. Such a role was further supported by Sequences 2 and 3 in real-time visualizations of the geniohyoid muscle contraction in human participant swallows through ultrasonography. Although the current study only aims to study obligatory sequences in healthy young adults, it is note-worthy at this point that these pairs are not obligatory (i.e., adherence below 100%) in two other groups (see middle and bottom heatmaps for healthy older adults and dysphagic adults, respectively in Appendix B). This somewhat resonates with the partially dissociative patterns of geniohyoid muscle contraction and hyoid bone displacement discussed in Section 4.1. Further investigation on the underlying causes of such patterns (e.g., sarcopenia and muscle control) is warranted.

While not identified as obligatory in the current study by an MLCS, high adherence was exhibited in two other paired events, namely (i) hyoid displacement onset, followed by maximum tongue base retraction (99.98% adherence), and (ii) maximum tongue base retraction followed by hyoid bone displacement offset (95.29% adherence). Given that 336 possible unique sequences were considered, this further supports the previous findings that, to achieve both swallowing safety and efficiency, airway protection and the upper esophageal sphincter (UES) opening shall always begin before the maximum bolus propulsion is reached, and the airway shall remain closed and the UES opened while the maximum propulsion of the bolus is taking place [16]. Similar to the obligatory sequences, these pairs show lower adherence, particularly the pair tongue base retraction maximum prior to hyoid bone displacement offset, among both healthy older adults and dysphagic adults.

With respect to the aim of the study, the conceptualization of encoding swallowing sequences as pseudo-DNA sequences permits the application of (M)LCS algorithms, which are potentially valid and efficient techniques for revealing (ab)normal swallowing physiology, as described in terms of the obligatory sequence (MLCS) and probability of event pair order (LCS) in both research and clinical contexts.

### 4.4. Limitations and Future Research

A few limitations should be noted in the present study. First, only a small sample of healthy young, healthy older, and dysphagic (with heterogenous and imbalanced etiologies) individuals were studied, and hence the results and proposals here require further evidence from larger scale studies, including more diverse and larger disordered group samples. Second, identical timestamps may indicate that a frame rate of 15 Hz might be inadequate to capture delicate and minute difference in swallowing event states, and hence future research shall consider a higher frame rate, while, in the long run, a desirable range of frame rate shall also be established. Since the current study only focused on swallowing event sequences of three key landmarks, the inclusion of more possible structures/events (e.g., hyolaryngeal approximation) may capture more information about the physiology of different types of swallowing. Also, while the current study mainly focused on gauging the effect of aging and dysphagia on the quantified variables, future studies may also investigate the effect of bolus volume and consistency on such variables among different subject groups. Additionally, given a global quantitative metric of temporality as an additional dimension of examining the swallowing process, it would be interesting to examine the correlation of such dimensions with kinematic (e.g., velocity and other spatio-temporal parameters) and morphological (e.g., cross sectional area and quality of muscles) measurements, particularly for geniohyoid and other suprahyoid muscles among healthy adults, as well as the effect of different conditions, such as sarcopenia, on an inventory of all these measurements. Last but not least, as the current study has provided preliminary support for sequence analyses as a promising feature extraction technique for POCUS swallowing examinations, manual annotation of USIs has remained labor-intensive. In this regard, future studies may also leverage machine learning techniques for automated tracking (e.g., action recognition or segmentation from timeseries) of swallowing movements to improve the reliability and efficiency of timestamp identification. This could also be coupled with sequence analyses on a larger sample size for validating screening/diagnostic accuracy so that the proposed conceptual framework and analytical methods could potentially be translated to clinical practice.

## 5. Conclusions

Ultrasonography is a promising non-invasive and accessible imaging modality that allows clinicians to examine the musculoskeletal properties and physiology of swallowing across different settings and various populations, particularly for the aging and frail population in the primary care, outreaching, and telehealthcare settings. Sequence analyses of swallowing movements in USIs showed that normal deglutition is characterized by intricate coordination of musculoskeletal control and highly concurrent and varying swallowing sequences to achieve swallowing safety and efficiency, which might be weakened or compromised by the possible effect of aged-related muscle changes and various conditions associated with dysphagia. Four obligatory swallowing sequences exclusive to healthy young adults were identified by the MLCS algorithm, and three of them involve geniohyoid muscle contraction, further highlighting the importance of the suprahyoid muscles to the physiology of normal deglutition. These provided preliminary empirical support for using an inventory of sequence analyses to extract quantitative, explainable, and clinically significant temporal features from USIs. As such, the study has also shown the plausibility of the systemic conceptualization of swallowing as instantiated sequences with concurrent subprocesses from a biological system. Despite the promising results, it should be noted that these newly proposed conceptualizations and measurement techniques shall further be applied on a larger and diverse population and validated against the current diagnostic gold standard, e.g., VFSS and FEES. Arguably, the conceptual and methodological proposals here are potentially applicable to other modalities that are capable of revealing swallowing event sequences, i.e., modality-independent. Furthermore, the proposed sequence analytical methods, which could be further automated with machine learning, may also serve as outcome measures for rehabilitation, provided that further studies verify the present findings.

## Figures and Tables

**Figure 1 jcm-13-02288-f001:**
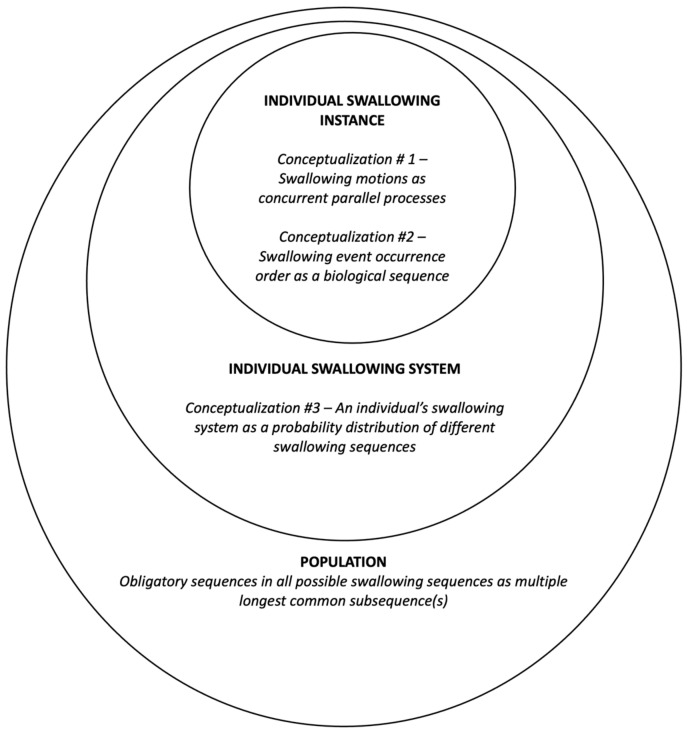
A systemic conceptualization of swallowing physiology at multiple levels.

**Figure 2 jcm-13-02288-f002:**
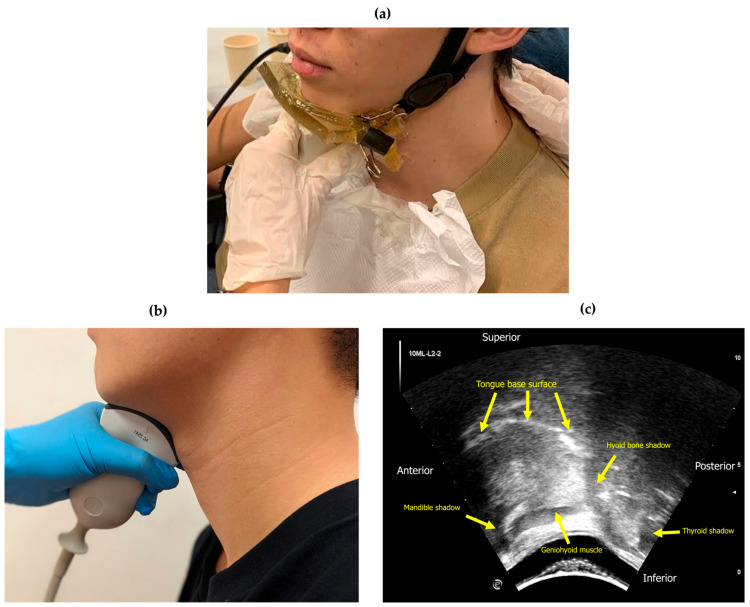
(**a**) Placement of the gel pad. (**b**) Placement of the transducer. (**c**) Annotated frame sample of USI acquired at the submental region on the mid-sagittal plane. Structures of interest: the surface of tongue base shown as a hyperechoic contour, hyoid bone as a reflective, acoustic shadow, and the geniohyoid muscle as a hypoechoic polygon.

**Figure 3 jcm-13-02288-f003:**
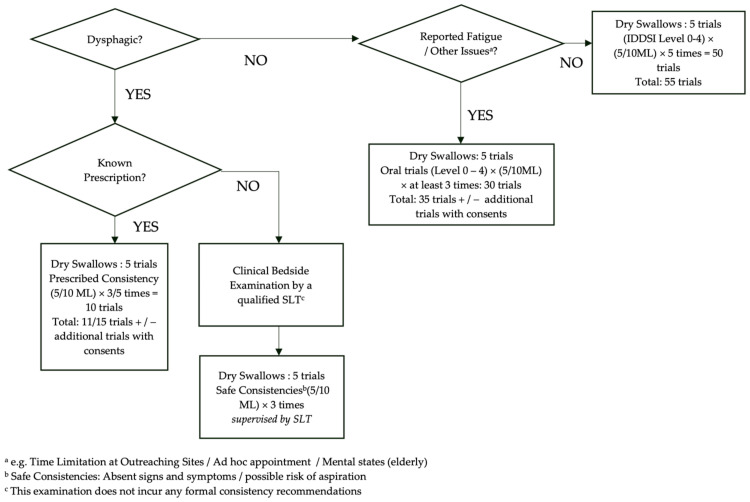
The Swallow Trial Protocol.

**Figure 4 jcm-13-02288-f004:**
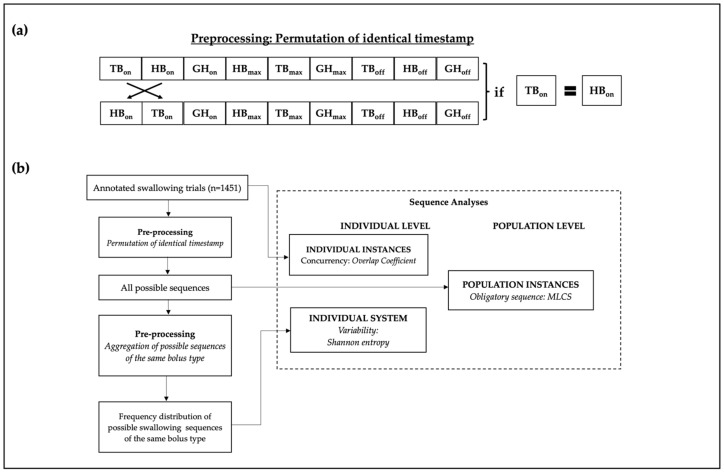
(**a**) A visual presentation of permuting identical timestamps. (**b**) Schematic representation of data extraction and analysis workflow.

**Figure 5 jcm-13-02288-f005:**
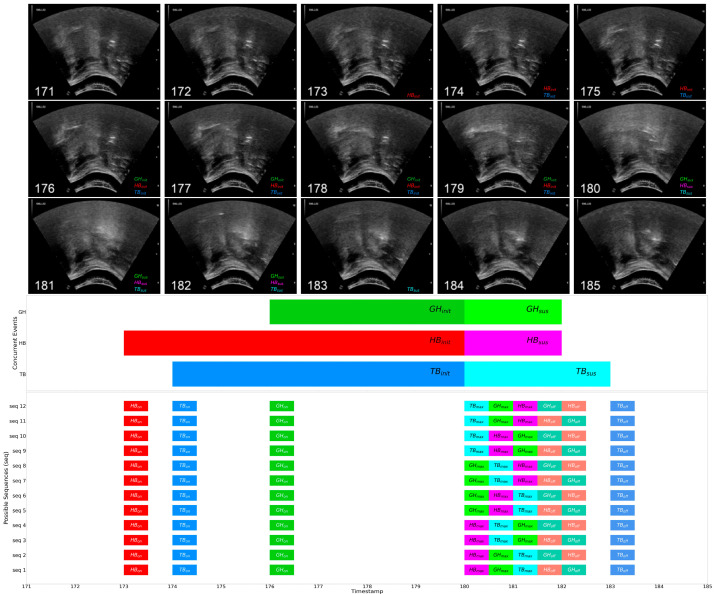
Annotated and extracted video frame sample from a healthy young participant (**top**), the corresponding concurrent event timelines in the form of a Gantt chart (**middle**), and the permuted possible event sequences (**bottom**).

**Figure 6 jcm-13-02288-f006:**
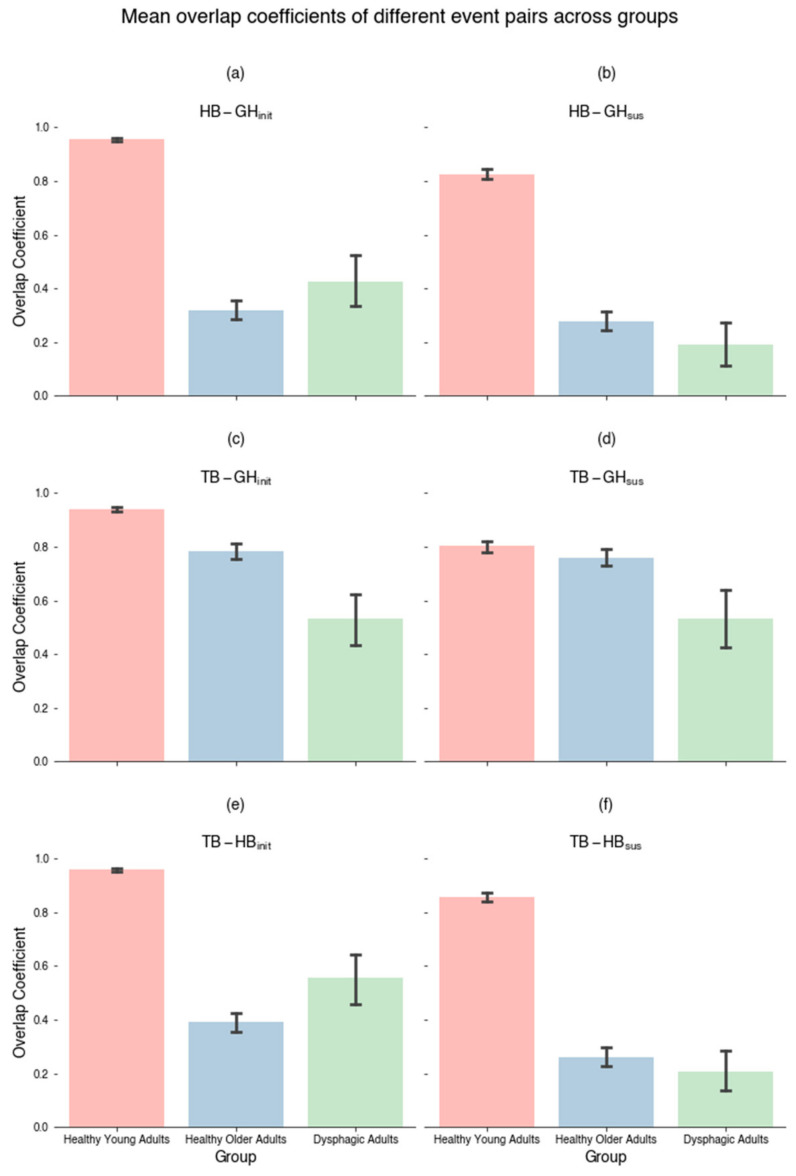
Mean overlap coefficients of different event transition pairs (error bars showing 95% Confidence Interval) across groups for overlapping (**a**) hyoid bone displacement and geniohyoid contraction initiation (HB-GH_init_), (**b**) sustained hyoid bone displacement and geniohyoid contraction (HB-GH_sus_), (**c**) tongue base retraction and geniohyoid contraction initiation (TB-GH_init_), (**d**) sustained tongue base retraction and geniohyoid contraction (TB-GH_sus_), (**e**) tongue base retraction and hyoid bone displacement initiation (TB-HB_init_), and (**f**) sustained tongue base retraction and hyoid bone displacement (TB-HB_sus_).

**Figure 7 jcm-13-02288-f007:**
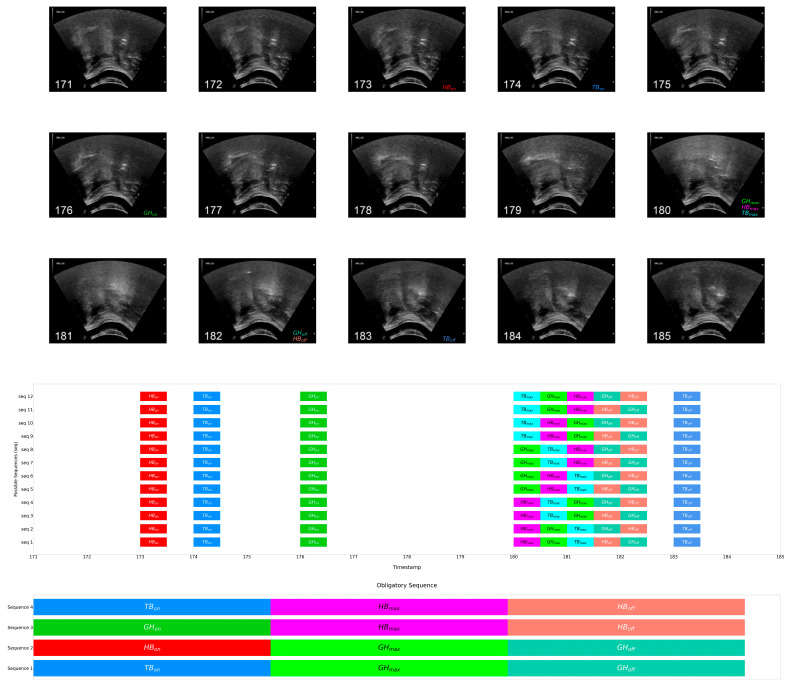
Video frame extracts annotated with event labels (**top**), possible sequences after permutation (**middle**), obligatory sequences found in the healthy young population (**bottom**).

**Table 1 jcm-13-02288-t001:** Demographic information of the participants.

Group	Mean Age in Years (SD)	Gender (M/F)	Etiology
Healthy Young Adult (*n* = 17)	41.18 (17.37)	7/10	-
Healthy Older Adult (*n* = 15)	69.46 (5.45)	6/9	-
Dysphagic Adult (*n* = 11)	63.55 (12.32)	4/7	neurological (e.g., stroke): 8 structural (e.g., head and neck cancer): 3

**Table 2 jcm-13-02288-t002:** Means, standard deviations (SD), results of a Kruskal–Wallis *H* Test, and post hoc Dunn’s test on the overlap coefficients of different event transition pairs across groups.

Event Transition Pair		Y	O	D	df	*H*	*p*-Value	*η* ^2^		Dunn’s Test	
		Mean (SD)	Mean (SD)	Mean (SD)					Y vs. O *p*-Value	Y vs. D *p*-Value	O vs. D *p*-Value
Initiating	HB-GH_init_	0.955 (0.088)	0.319 (0.428)	0.427 (0.416)	2	622.700	<0.0001 ***	0.429	<0.0001 ***	<0.0001 ***	0.598
	TB-GH_init_	0.938 (0.122)	0.785 (0.353)	0.533 (0.441)	2	99.907	<0.0001 ***	0.068	<0.0001 ***	<0.0001 ***	<0.0001 ***
	TB-HB_init_	0.959 (0.092)	0.391 (0.434)	0.556 (0.450)	2	526.413	<0.0001 ***	0.362	<0.0001 ***	<0.0001 ***	0.004 **
Sustaining	HB-GH_sus_	0.825 (0.263)	0.277 (0.439)	0.190 (0.364)	2	456.167	<0.0001 ***	0.314	<0.0001 ***	<0.0001 ***	0.175
	TB-GH_sus_	0.802 (0.287)	0.759 (0.373)	0.534 (0.485)	2	16.745	<0.001 ***	0.011	0.99	<0.001 **	<0.001 **
	TB-HB_sus_	0.856 (0.239)	0.260 (0.423)	0.207 (0.341)	2	571.850	<0.0001 ***	0.394	<0.0001 ***	<0.0001 ***	0.539

Note: Y = Healthy Young Adults. O = Healthy Older Adults. D = Dysphagic Adults. ** *p* < 0.01. *** *p* < 0.001.

**Table 3 jcm-13-02288-t003:** Means, standard deviations (SD), results of the Kruskal–Wallis *H* Test, and post hoc Dunn’s test on Shannon entropy on matched and all bolus types across groups.

Bolus Types	Y	O	D	df	*H*	*p*-Value	*η* ^2^		Dunn’s Test	
	Mean (SD)	Mean (SD)	Mean (SD)					Y vs. O *p*-Value	Y vs. D *p*-Value	O vs. D *p*-Value
Matched—Dry and 5 mL of IDDSI Levels 0–4	5.119 (1.095)	4.428 (0.950)	3.038 (0.872)	2	52.253	<0.0001 ***	0.260	<0.001 **	<0.0001 ***	<0.0001 ***
All bolus types	5.07 (1.075)	4.286 (1.095)	2.896 (1.011)	2	78.472	<0.0001 ***	0.224	<0.0001 ***	<0.0001 ***	<0.0001 ***

Note: Y = Healthy Young Adults. O = Healthy Older Adults. D = Dysphagic Adults. ** *p* < 0.01. *** *p* < 0.001.

**Table 4 jcm-13-02288-t004:** Event order probability in the healthy young adult groups.

	TB_on_	HB_on_	GH_on_	TB_max_	HB_max_	GH_max_	TB_off_	HB_off_	GH_off_
**TB_on_**		0.3838	0.4963	1.0	1.0 *	1.0 *	1.0	1.0	1.0
**HB_on_**			0.6162	0.9998 ^^^	1.0	1.0 *	1.0	1.0	1.0
**GH_on_**				0.9992	1.0 *	1.0	0.9998	1.0	1.0
**TB_max_**					0.4755	0.5181	1.0	0.9858 ^^^	0.9793
**HB_max_**						0.6105	0.9529	1.0 *	0.9733
**GH_max_**							0.9309	0.9917	1.0 *
**TB_off_**								0.4804	0.4904
**HB_off_**									0.5302
**GH_off_**									

Note: * identified as non-trivial obligatory sequence by the leveled DAG MLCS algorithm. ^ previously identified as obligatory pairs in [16].

## Data Availability

Supporting data can be provided by contacting with the corresponding author.

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
