# Peer review of "Using Sequence Analyses to Quantitatively Measure Oropharyngeal Swallowing Temporality in Point-of-Care Ultrasound Examinations: A Pilot Study"

_jcm, 2024, doi:10.3390/jcm13082288_

Round 1

Reviewer 1 Report

Comments and Suggestions for Authors

In this manuscript (jcm-2917611), the authors tried to quantitatively measure oropharyngeal swallowing using sequence analyses based on point-of-care ultrasound swallowing examinations. They showed that the concurrency of movement sequence and swallowing sequence variability were impacted by aging and dysphagia and identified four obligatory sequences. However, this MS may be too difficult to understand overall even for clinicians who engage in dysphagia rehabilitation:

1.      Figures 1 and 4 are incomprehensible only on first impressions and need to be simplified.

2.      The information on 11 dysphagic adults is inadequate. In which phase of the oral or pharyngeal stage did they have swallowing impairments? Didn’t it have an impact on the results of point-of-care ultrasound swallowing examinations?

3.      I could not understand lines 212-214 on page 7, “a short hum (/m/) was performed before and after each the swallow for quick reference to the Frames of Interest (FOIs) as well as using vocal fold vibrations as a landmark for relative referencing to the Structures of Interest (SOIs)”. It requires a supplementary explanation.

4.      At least, “HB-GHinit”, “HB-GHsus”, and “TB-HBsus” need to be graphically illustrated. Without any diagrammatic representation, it may be difficult for many readers to create distinct images of them.

5.      The authors cited literature on multiple Longest Common Subsequence (MLCS) algorithms, which also require a supplementary explanation. Many readers may have not the remotest of MLCS.

6.      Sequences 1-4 also need to be graphically illustrated. Without any diagrammatic representation, it may be also difficult for many readers to create distinct images of them. While Sequences 1-4 appeared on page 13 in Results using abbreviated words, they did on page 17 in Discussion using words spelled out.

7.      No difference between Figures 6 and 7 may be conveyed to readers. Figure 8 also did not have much graphical effect. They seem to be enough to be presented as tables.

8.      In line 250 on page 8, “and” appeared repeatedly.

9.      In Abstract, the word “POCUS” abbreviated in line 31 and spelled out in 35 appeared in reverse order.

Reviewer 2 Report

Comments and Suggestions for Authors

This is a very interesting study about the possibility of using a new protocol with ultrasound (US) to explore swallowing.

  1. Keywords: I propose using Mesh keywords.

Introduction: In the first paragraph of section 1.1, I suggest adding that protocols have been proposed to screen swallowing more effectively, especially in neurological disorders, considering submental, pharyngeal, and laryngeal structures (Chiaramonte R, Di Luciano C, Chiaramonte I, Serra A, Bonfiglio M. Multi-disciplinary clinical protocol for the diagnosis of bulbar amyotrophic lateral sclerosis. Acta Otorrinolaringol Esp (Engl Ed). 2019 Jan-Feb;70(1):25-31. English, Spanish. doi: 10.1016/j.otorri.2017.12.002; Diaz K, Stegemöller EEL. Electromyographic measures of asymmetric muscle control of swallowing in Parkinson's disease. PLoS One. 2022 Feb 18;17(2):e0262424. doi: 10.1371/journal.pone.0262424.).

Results: It might be beneficial to include a video in the results section showing the ultrasound movement of the involved structures, along with an explanation of the proposed sequences.

Discussion: At the beginning of the discussion, I recommend providing a clearer and simpler explanation of the statistical results, summarizing them more effectively.

In the conclusion, emphasize again the need to acknowledge that, as this is a new proposed method of diagnosis, many trials with numerous patients are required to validate its efficacy fully.

Comments on the Quality of English Language

 Minor editing of English language required

Round 2

Reviewer 1 Report

Comments and Suggestions for Authors

In this manuscript (jcm-2917611), the authors revised the first version. There are still some difficult parts in figures and explanations to interpret, but the authors tried various measures to make them more approachable using Supplementary Materials together.

These revisions may be accessible to readers.

On the other hand, the blue and green colors of the words, “TBinitand GHinitin the video frames of Figures 5 and 7 are very hard to see.

Reviewer 2 Report

Comments and Suggestions for Authors

I thanks you for your effort in modifying the article. I appreciate the originality and now the fluid readibility. 
